# Essential roles of Lon protease in the morpho-physiological traits of the rice pathogen *Burkholderia glumae*

**Eunhye Goo**[1,2]*, **Ingyu Hwang**[1,2]

**1** Department of Agricultural Biotechnology, Seoul National University, Seoul, Republic of Korea, **2** Research Institute of Agriculture and Life Sciences, Seoul National University, Seoul, Republic of Korea

\* sourire11@snu.ac.kr

## Abstract

The highly conserved ATP-dependent Lon protease plays important roles in diverse biological processes. The *lon* gene is usually nonessential for viability; however, *lon* mutants of several bacterial species, although viable, exhibit cellular defects. Here, we show that a lack of Lon protease causes pleiotropic effects in the rice pathogen *Burkholderia glumae*. The null mutation of *lon* produced three colony types, big (BLONB), normal (BLONN), and small (BLONS), in Luria–Bertani (LB) medium. Colonies of the BLONB and BLONN types were re-segregated upon subculture, while those of the BLONS type were too small to manipulate. The BLONN type was chosen for further studies, as only this type was fully genetically complemented. BLONN-type cells did not reach the maximum growth capacity, and their population decreased drastically after the stationary phase in LB medium. BLONN-type cells were defective in the biosynthesis of quorum sensing (QS) signals and exhibited reduced oxalate biosynthetic activity, causing environmental alkaline toxicity and population collapse. Addition of excessive *N*-octanoyl-homoserine lactone (C8-HSL) to BLONN-type cell cultures did not fully restore oxalate biosynthesis, suggesting that the decrease in oxalate biosynthesis in BLONN-type cells was not due to insufficient C8-HSL. Co-expression of *lon* and *tofR* in *Escherichia coli* suggested that Lon negatively affects the TofR level in a C8-HSL-dependent manner. Lon protease interacted with the oxalate biosynthetic enzymes, ObcA and ObcB, indicating potential roles for the oxalate biosynthetic activity. These results suggest that Lon protease influences colony morphology, growth, QS system, and oxalate biosynthesis in *B. glumae*.

## Introduction

The ATP-dependent protease Lon is highly conserved in prokaryotes and eukaryotes and is involved in diverse biological processes [1]. Lon was first identified in *Escherichia coli* and derived its name from the long, non-septate filamentation phenotype of the *lon* mutant [2]. Since its discovery, studies have shown that Lon plays an important role in the degradation of abnormal or unstable regulatory proteins as a quality control protease [1]. In *E. coli*, the short-

**Data Availability Statement:** All relevant data are within the paper and its Supporting Information files.

**Funding:** This work was supported by Basic Science Research Program through the National

Research Foundation of Korea (NRF) funded by the Ministry of the Education (2021R1I1A1A01040314) (E.G.). The funders had no role in study design, data collection and analysis, decision to publish, or preparation of the manuscript.

**Competing interests:** The authors have declared that no competing interests exist.

lived regulatory proteins selectively degraded by Lon include bacteriophage λ N protein, the cell division regulator SulA, the positive regulator of capsule synthesis, RcsA, and the F factor plasmid antidote protein CcdA [3–5]. The *lon* mutant of *E. coli* exhibited increased sensitivity to UV irradiation, filamentation, and overproduction of capsular polysaccharides [3–5]. In *Caulobacter crescentus*, Lon protease is involved in regulating the methylation of chromosomal DNA and cellular differentiation via degradation of cell-cycle-regulated DNA methyltransferase (CcrM) [6]. In pathogenic bacteria such as *Brucella abortus* and *Salmonella typhimurium*, Lon-mediated proteolysis of transcription factors controlling for the expression of virulence genes is necessary for host invasion [7, 8]. Lon protease influenced the biofilm formation, motility, and survival of *Pseudomonas aeruginosa* in a rat model of chronic infection [9].

The *lon* gene is generally nonessential for viability. However, LonV is essential for viability, while LonD plays an important role in sporulation in *Mycobacterium xanthus* [10, 11]. A recent study showed that Lon protease is essential for growth under stressful conditions in pathogenic bacteria such as *Acinetobacter baumannii* and *Dickeya solani* [12, 13]. Lon protease affects cell morphology in *Agrobacterium tumefaciens* [14]. A *lon* mutant in *A. tumefaciens* formed both large and small colonies, of which the large colonies formed homogeneously sized colonies after subculture, while the small colonies re-differentiated into small and large colonies after subculture [14].

Lon protease has been reported to play a negative regulatory role in the quorum sensing (QS) system of *P. aeruginosa* and *P. putida*, but the underlying molecular mechanisms have not yet been determined [15, 16]. We aimed to identify the roles of Lon protease in QS-dependent phenotypes including phytotoxin production, motility, protein secretion, and oxalate biosynthesis in *Burkholderia glumae*, the causal agent of rice panicle blight [17–20]. We generated a *lon* null mutant of *B. glumae* to assess its roles in QS and found that the mutant exhibited continuous segregation of cell morphology in the mutant upon subculture. Such segregation was an obstacle in deciphering the functional roles of Lon protease. Nonetheless, we selected one genetically complementary colony type from among the three different colony types present and analyzed the effect of Lon protease on the QS system and QS-dependent phenotypes. We found that defects in the Lon protease resulted in pleiotropic phenotypes, including difficulty reaching maximum growth capacity, failure to generate QS signals, and alkalization of the growth environment, in addition to giving colony size variations. Our findings provide insights into the critical functions of Lon protease in *B. glumae*, including maintenance of cellular integrity and roles in the QS system and the QS-dependent phenotypes.

## Materials and methods

### Bacterial strains and growth conditions

The bacterial strains and plasmids used in this study are listed in S1 Table. Strains of *B. glumae* and *E. coli* were grown in Luria-Bertani (LB) broth (Affymetrix) (0.1% tryptone, 0.5% yeast extract, and 0.5% NaCl [all wt/vol]) at 37°C, M9 minimal medium ($Na_2HPO_4$, $KH_2PO_4$, NaCl, and $NH_4Cl$) with 0.2% glucose, nutrient broth (NB) medium (BD) (0.1% beef extract, 0.2% yeast extract, 0.5% peptone and 0.5% NaCl [all wt/vol]), and King's B medium (BD) (2% peptone, 0.15% $K_2HPO_4$, 0.15% $MgSO_4$ [all wt/vol]) with the following antibiotics: gentamycin, 20 μg/mL; rifampicin, 100 μg/mL; tetracycline, 10 μg/mL; ampicillin, 50 μg/mL; trimethoprim; 75 μg/mL; kanamycin, 50 μg/mL.

### Cell viability and extracellular pH assay

Cells were inoculated in 2 mL LB broth with appropriate antibiotics and grown at 37°C at 250 rpm for 18 h. Overnight cultures were washed twice with fresh LB broth, and turbidity was

adjusted to an optical density (OD) of 0.05 at 600 nm using a BioSpectrometer (Eppendorf) followed by 2 mL subculture in glass test tubes (PYREX) for all assays. To compare initial growth rate of wild type and mutant, an $OD_{600}$ of 0.05 was diluted 20 times followed by subculture (Figs 2A and S1A). Aliquots of 100 μL from each sample were serially diluted and 10 μL each of three repeats was spotted on LB agar medium to monitor colony-forming units (CFUs) at the designated time point. LB agar plates were incubated at 37°C for 24 h to allow colonies to grow. CFUs were counted under a dissecting microscope and multiplied by the appropriate dilution factor to calculate CFU/mL. In an extracellular pH assay, the culture supernatant was sampled from each vial at the designated time point and the pH was measured using a pH meter (Lab 860, SCOTT Instruments).

## Construction of the *B. glumae lon*::Gm, *lon*::Gm/*lon*, *tofI*::Sp/*lon*::Gm double mutant, and *obcA*::Tp

The *lon* (bglu_1g13520) gene was deleted via insertion of the gentamycin cassette *Sca*I and marker exchange as described previously [21]. The pLa2 is a 9.0-kb *EcoR*I-*Hind*III fragment cloned into pLAFR3, including the *lon* gene from cosmid clone pLa1. A gentamycin cassette was generated by PCR using ScaI-Gm-F and Gm-ScaI-R primers, cloned into pBluescript II SK(+), and termed "pBS_Gm4". A 0.8-kb DNA fragment obtained from pBS_Gm4 via digestion performed with *Sca*I was ligated into pLa2 to produce pLa2_Gm. pLa2_Gm was mobilized from DH5α into *B. glumae* BGR1 or BGS2 by conjugation using pRK2013 as a helper plasmid and marker exchange, as described previously [21]. For genetic complementation of the *lon* mutant, a 2.876-kb DNA fragment containing the *lon* gene and its own promoter was generated by PCR using HindIII-La and La-EcoRI primers, cloned into pBluescript II SK(+), and termed "pLa3". A 2.876-kb DNA fragment obtained by pLa3 by digestion performed with *Hind*III and *EcoR*I was ligated into pLAFR3 to produce pLa4. pLa4 was mobilized from DH5α into *B. glumae* BLONN through conjugation using pRK2013 as a helper plasmid. The *obcA* (bglu_2g18790) gene was deleted via insertion of the trimethoprim resistance gene *Kpn*I-*Xho*I and marker exchange as described previously [21]. A 2.716-kb DNA fragment including the *obcA* and *obcB* genes and their promoter was generated via PCR using BamHI-obcA and obcB-EcoRI primers, cloned into pLAFR3, and termed "pOBC4". pTP2 contained a trimethoprim-resistant gene produced by PCR using the Tp(BamHI)-F and Tp-R primers and then cloned into pBluescript II SK(+). A 0.887-kb *Kpn*I-*Xho*I fragment from pTP2 was cloned into pOBC4 to produce pOBC4::Tp. The BOBCA strain was generated via triparental mating of *B. glumae* BGR1, DH5α (pRK2013), and DH5α (pOBC4::Tp). Southern blot analysis was performed to confirm the BLONN, S2LON, and BOBCA strains, and the BLONC was assessed via plasmid extraction.

## Measurements of oxalate levels and oxalate biosynthetic activity assay

Oxalate levels and oxalate biosynthetic activity were measured as described previously [20, 22]. Oxalate was measured using an oxalate assay kit (Trinity Biotech) according to the manufacturer's instructions. Briefly, oxalate was converted into carbon dioxide and hydrogen peroxide using oxalate oxidase, and the production of hydrogen peroxide was measured via reaction with 3-(dimethylamino) benzoic acid, during which it formed a blue compound catalyzed by peroxidase. Absorbance at 590 nm was measured using a microplate reader (PerkinElmer); 0.5 mM oxalate was used as a standard. The absorbance of the sample was divided by the absorbance of standard and multiplied by the appropriate dilution factor. The reaction buffer used in the assay measuring oxalate biosynthetic activity consisted of 200 mM Tris-Cl (pH 8.0), 10 mM EDTA, 20 mM $CoCl_2$, 2 mM acetyl-CoA, and 200 mM oxaloacetate. The total cell lysate was added to the reaction mixture, followed by incubation at 37°C for 10 min. The level of

biosynthesized oxalate from the reaction was measured using an oxalate assay kit. Units were calculated as the concentration of biosynthesized oxalate divided by the amount of total protein and reaction time.

## RNA extraction and quantitative reverse-transcription PCR (qRT-PCR)

Total RNAs were extracted from *B. glumae* BGR1, BLONN (BGR1 *lon*::Gm), and BLONC (BGR1 *lon*::Gm/*lon*), grown in LB medium at 37˚C for 10 h after subculture, were extracted using RNeasy mini kits (Qiagen) following the manufacturer's protocols. The extracted total RNA was treated with RNase-free DNase I (Ambion) to remove DNA. Total RNA (1 μg) was reverse-transcribed into cDNA using M-MLV Reverse Transcriptase (Promega, Madison, WI, USA) and incubated for 1 h at 42˚C. The primer pairs used for qRT-PCR are listed in S2 Table. The 16S rRNA gene was used as the positive control. Transcription levels were determined using SsoFast EvaGreen Supermix (Bio-Rad, Hercules, CA, USA) and the CFX96 Real-Time PCR System (Bio-Rad). The thermal cycling parameters were as follows: 95˚C for 30 s, followed by 40 cycles of 95˚C for 5 s and 60˚C for 5 s. All reactions were performed in triplicate, and all data were normalized to the expression levels of the 16S rRNA gene expression using Bio-Rad CFX Manager software.

## Overexpression of Lon and TofR and co-expression of Lon and TofR

The *lon* and *tofR* genes of *B. glumae* were amplified using the primers listed in S2 Table. The amplification products were cloned into the *Nhe*I and *Hin*dIII restriction sites of the pET-28b expression vector and into the *Nde*I and *Xho*I restriction sites of the pACYC-Duet expression vector (Novagen), resulting in pLa8 and pTOFR5, respectively. His-Lon-His and TofR-His were overexpressed in *E. coli* BL21(DE3), which was induced by addition of 1 mM isopropyl β-o-thiogalactopyranoside (IPTG), followed by additional growth for 1 h at 37˚C. To test for protease resistance against Lon protease, 4 μM *N*-octanoyl-homoserine lactone (C8-HSL) was added to the BL21(DE3) cultures. The BL21(DE3) carrying both pLa8 and pTOFR5 was used for co-expression of Lon and TofR.

## Immunoblotting using anti-ObcA, anti-ObcB, anti-TofR, anti-HA, and anti-His antibodies

Total lysates from the *B. glumae* and *E. coli* strains were separated by 10% sodium dodecyl sulfate-polyacrylamide gel electrophoresis (SDS-PAGE) and transferred to polyvinylidene fluoride (PVDF) membranes. The membranes were blocked using 4% skim milk and incubated with anti-ObcA (1:4000) plus anti-ObcB (1:3000), anti-TofR (1:3000), anti-HA (Invitrogen, 1:2500), and anti-His (Invitrogen, 1:3000) antibodies. Proteins were detected using a secondary anti-rabbit horseradish peroxidase-conjugated IgG antibody (Cell Signaling), a secondary anti-mouse horseradish peroxidase-conjugated IgG antibody (Cell Signaling), and a chemiluminescent substrate (Bio-Rad). Antibodies against *B. glumae* ObcA, ObcB, Lon, and TofR were generated in rabbits immunized with ObcA-6xHis, a peptide of ObcB (N-GNIEFYA DQRRPQYLRELVRVTR-C), a peptide of Lon (N-GLPWRKKSKVNNDLSNAE-C), and TofR-6xHis. The density (pixels/inch) of each band was measured using ImageJ version 1.53a software (NIH). All experiments were performed using three independent replicates.

## Coimmunoprecipitation (Co-IP) and western blot analysis

According to the manufacturer's instructions, Co-IP was performed using the Pierce Direct IP kit (Thermo Fisher Scientific). The anti-ObcA (10 μg), anti-ObcB (10 μg), and anti-Lon (10 μg)

antibodies were coupled to AminoLink Plus Coupling Resin (Thermo Fisher Scientific). Cell lysates from *B. glumae* BGR1 and S2HA containing 1 mg of protein were added to the antibody-coupled resin in a spin column and mixed at 4°C for 24 h. The immunoprecipitates were eluted, followed by the addition of a 5× non-reducing lane marker sample buffer. Samples were boiled for 5 min, cooled to room temperature, and separated via SDS-PAGE. The samples were then transferred to PVDF membranes. Anti-Lon, anti-ObcA, anti-ObcB, and anti-HA (Invitrogen) antibodies were used to detect interactions of ObcA with ObcB and Lon, of ObcB with Lon, and of TofI-HA with Lon, respectively. The immunoreactive bands were detected using electrochemiluminescence (ECL) reagents (Bio-Rad) and captured using the ChemiDoc™ XRS+ imager (Bio-Rad). All experiments were performed using at least three independent replicates.

## Autoinducer assay

An autoinducer assay was performed as described previously [18] with the following modifications; *B. glumae* BGR1, BLONN (BGR1 *lon*::Gm), and BLONC (BGR1 *lon*::Gm/*lon*) were grown in LB supplemented with HEPES (pH 7.0) for 24 h. Autoinducer was extracted from the 500 μL culture-free supernatant using the same amount of ethyl acetate and dried. The dried pellet was dissolved in 10 μL dimethyl sulfoxide. Autoinducer was developed via the thin layer chromatography (TLC), using 70% methanol in the TLC tank. LB agar with biosensor strain *Chromobacterium violaceum* CV026 overlaid was used to develop the TLC plate to visualize C8-HSL and C6-HSL.

## Statistical analysis

Statistical analyses were performed using SPSS version 23 (IBM Corp.). Comparisons were performed using one-way analysis of variance (ANOVA) followed by Tukey's post-hoc analysis (set at 5% significance level).

# Results

## Continuous segregation of the *lon* mutant into morphologically distinct colonies

To explore the role of Lon protease in *B. glumae* BGR1, we constructed a *lon* null mutant by inserting a gentamicin resistance cassette into the *lon* gene via double-crossover homologous recombination. The genome of *B. glumae* BGR1 contains three homologs of ATP-dependent Lon protease: bglu_1g13520, bglu_1g31260, and bglu_1g33380. The bglu_1g13520 homolog had typical domains of Lon protease, including the Lon substrate binding domain (accession number_pfam02190), P-loop_NTPase domain (accession number_ cl38936), and C-terminal proteolytic domain (accession number_pfam05362) (S1 Fig). On the other hand, bglu_1g31260 and bglu_1g33380 contained only N-terminal domain of Lon protease (accession number_cl19481) and P-loop_NTPase domain, respectively (S1 Fig). The bglu_1g13520 was selected for investigation of its biological roles in *B. glumae* BGR1. The marker-exchanged *lon* null mutant was segregated into three morphologically distinct colony types: big (BLONB), normal (BLONN), and small (BLONS) (Fig 1A). The BLONS-type cells grew very slowly and were too small to manipulate. The largest-sized colonies, called BLONB, grew faster than the other two types but segregated into BLONB and BLONS only, not BLONN, after subculture in LB medium (Fig 1B). BLONB was bigger than BLONN in all tested media, and was darker, than BLONN under a dissecting microscope (Fig 1A, 1B and 1E–1H). The occurrence frequency of BLONB and BLONS segregated from BLONB was 67.5% and 32.5%, respectively (Fig 1C). When a colony of the BLONN type was subcultured in LB medium, cells were segregated into the BLONB, BLONN,

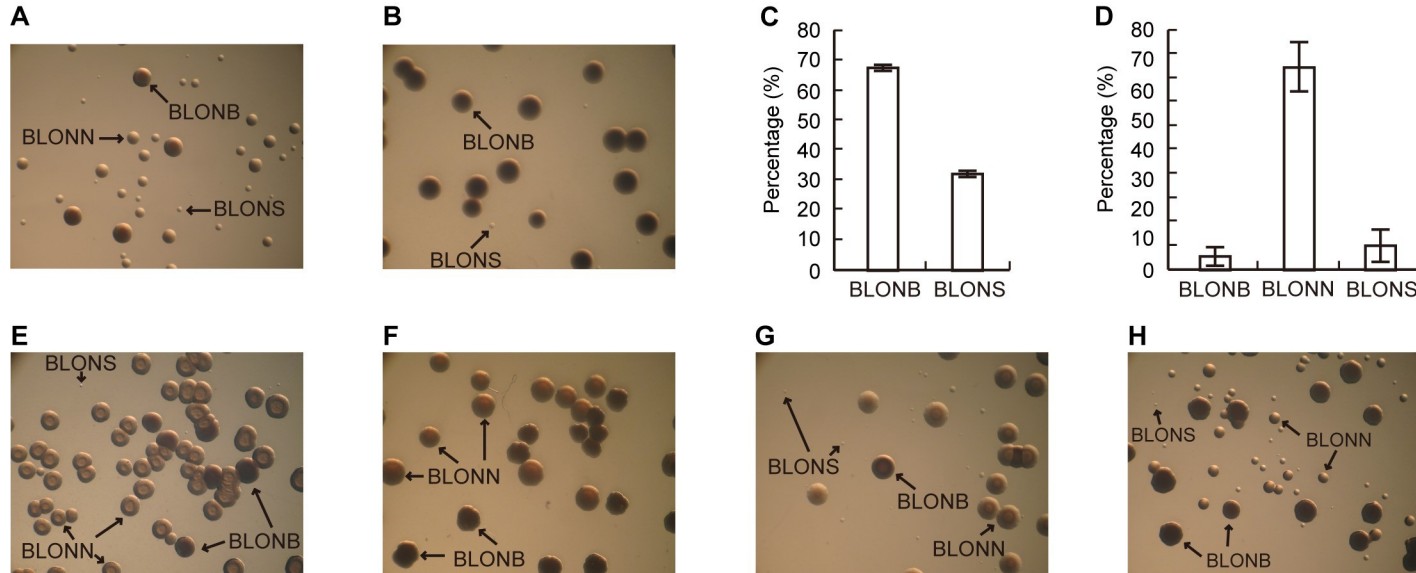

**Fig 1. Segregation of morphologically distinct colony types from the *lon* mutant.** (A) The *lon* mutant was segregated into three morphologically distinct colony types, big (BLONB), normal (BLONN), and small (BLONS). Colonies were grown in LB medium and observed under a dissecting microscope at ×30 magnification. (B) The BLONB segregated into BLONB and BLONS after subculture in LB medium. Colonies were observed under a dissecting microscope at ×30 magnification. (C) The BLONB and BLONS that had segregated from BLONB subcultured in LB medium accounted for 67.5% and 32.5% of the total population, respectively. (D) When the BLONN was subcultured in LB medium, 5.35%, 84.52%, and 10.13% of cells segregated into BLONB, BLONN, and BLONS types, respectively. The colony morphology of BLONN subcultured in (E) M9 minimal medium supplemented with 0.2% glucose, (F) nutrient broth medium, (G) King's B medium, (H) LB supplemented with 100 mM HEPES (pH 7.0).

and BLONS types at frequencies of 5.35%, 84.52%, and 10.13%, respectively (Fig 1D). Such segregation of the BLONB and BLONN types did not change in M9 minimal medium supplemented with 0.2% glucose (Fig 1E), nutrient broth medium (Fig 1F), King's B medium (Fig 1G), and LB supplemented with 100 mM HEPES (pH 7.0) (Fig 1H). We selected the BLONN type for further analysis, as this colony type exhibited fully complemented phenotypes with pLa4 carrying a full length of *lon* gene along with its own promoter in pLAFR3 (Fig 2A).

## Impaired growth of the *lon* mutant

To determine the biological significance of Lon protease, we measured the growth and extracellular pH of the *lon* mutant. The population of the wild type BGR1 reached a maximum capacity at $3.20 \times 10^9$ colony forming unit (CFU)/mL after 12 hours of subculture in LB medium (Fig 2B). On the other hand, the *lon* mutant increased up to $2.26 \times 10^8$ CFU/ml after the stationary phase and decreased gradually for 1 to 3 days after subculture, followed by complete population death (Fig 2B). During the initial growth stage, the *lon* mutant had a significantly lower growth rate than that of the wild type (Fig 2A). The *lon* mutant showed alkalization of the LB culture medium, whereas the pH of wild-type BGR1 slightly increased to 7.33 and then decreased to 4.97, and finally returned to 6.67 (Fig 2C). Environmental pH increased during growth of the *lon* mutant, which resulted in a population crash (Fig 2). Addition of 100 mM HEPES buffer to the growth medium spared the *lon* mutant from cell death after the stationary phase but did not affect its initial growth rate (S2 Fig).

## Failure of QS signal biosynthesis in the *lon* mutant

To determine whether impaired growth of the *lon* mutant affects QS systems, we performed an autoinducer (AI) assay of the wild type and the *lon* mutant. We performed AI assays with

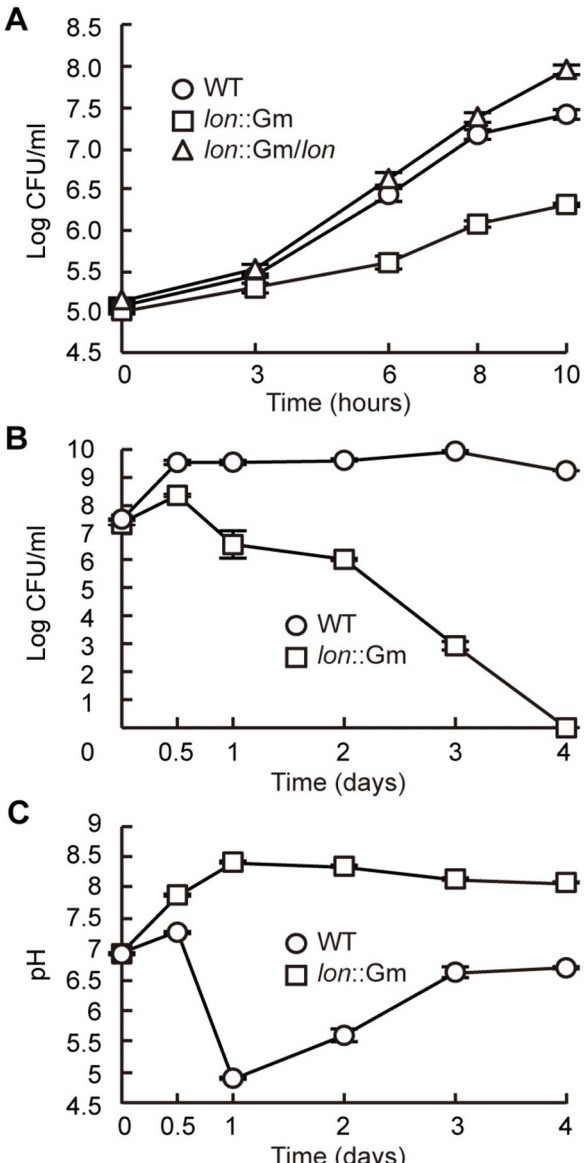

**Fig 2. Cell viability and extracellular pH of the *lon* mutant of *B. glumae*.** (A) *B. glumae* strains were subcultured in LB medium with 20× dilution after adjusting samples to an O.D$_{600}$ of 0.05. CFUs were determined from serial dilution of 100 μL aliquots of each sample at the designated time points. The *lon* mutant had a significantly lower initial growth rate than that of the wild type. (B) *B. glumae* strains were subcultured in LB medium following adjustment of samples to an O.D$_{600}$ of 0.05. The population density of the *lon* mutant decreased gradually after 12 h, leading to a population crash. (C) The extracellular pH of the strains was measured at each time point using a pH meter.

wild type BGR1 at two different population densities, approximately at $1.28 \times 10^8$ and $1.33 \times 10^9$ CFU/mL, as the *lon* mutant did not reach the maximum population density. AI signals were detected at $1.28 \times 10^8$ CFU/mL but much higher levels were observed at $1.33 \times 10^9$ CFU/mL of wild type BGR1 (Fig 3A). However, the *lon* mutant produced no detectable QS signals, and the genetic complementation resulted in recovery of QS signal biosynthesis to the wild type level (Fig 3A). As the failure of the *lon* mutant to reach maximum growth capacity might be one reason that the mutant did not produce detectable amounts of QS signals, we harvested cells from 10 culture tubes, each containing $3.57 \times 10^8$ CFU/mL, and pooled them to

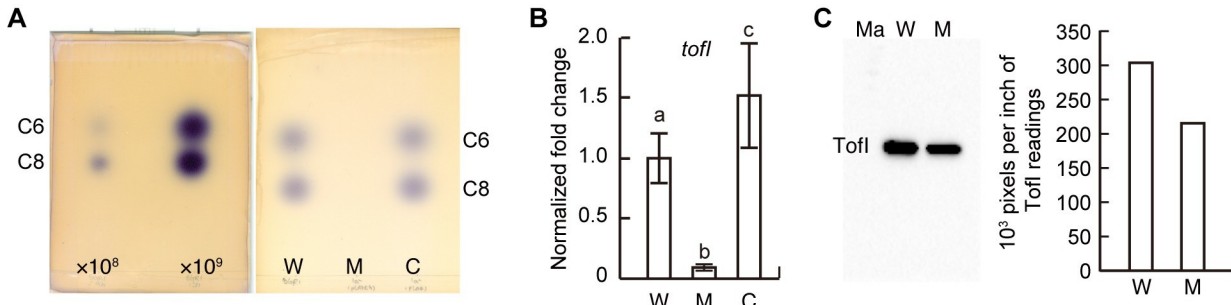

**Fig 3. Production of QS signals by the wild type and the complemented strains but not the *lon* mutant.** (A) A significantly larger amount of QS signal was produced by wild-type cells of $1.33 \times 10^9$ CFU/mL than the wild-type cells of $1.28 \times 10^8$ CFU/mL. The labels $\times 10^8$ and $\times 10^9$ denote the wild-type cell densities of $1.28 \times 10^8$ CFU/mL and $1.33 \times 10^9$ CFU/mL. TLC analysis of acyl-HSLs extracted from the wild type ($3.48 \times 10^9$ CFU/mL), the *lon* mutant ($7.08 \times 10^8$ CFU/mL), and the complemented strain ($3.03 \times 10^9$ CFU/mL) grown for 24 h in LB supplemented with 100 mM HEPES (pH 7.0). The C6-HSL and C8-HSL were visualized using the acyl-HSL sensor strain CV026. C6, C8, W, M, and C denote C6-HSL, C8-HSL, the wild type, *lon* mutant, and complemented strain, respectively. (B) Expression of the *tofI* gene in each strain was determined during mid-log phase by qRT-PCR. W, M, and C denote the wild type, *lon* mutant, and complemented strains, respectively. The letters (a, b, and c) above each mean indicate significant differences based on a one-way ANOVA, followed by Tukey's post-hoc analysis. A value of $p < 0.05$ indicates significant differences among the strains. (C) The amount of TofI-HA in the wild type and *lon* mutant during early stationary phase was determined via western blot analysis using an anti-HA antibody. The densities (pixels/inch) of the TofI-HA bands are presented alongside the blot. W and M denote the wild type and *lon* mutant, respectively.

reach approximately $3.57 \times 10^9$ CFU/mL. Pooled samples were cultured for an additional 24 hours in the LB supplemented with 100mM HEPES (pH 7.0) prior to QS signal assays (S3 Fig). No QS signal was detected at $3.57 \times 10^9$ CFU/mL of the *lon* mutant (S3 Fig). To determine whether the lack of QS signals in the *lon* mutant was due to low expression of QS signal synthase gene, *tofI*, we assessed *tofI* expression at the transcription level during the mid-log phase using qRT-PCR. Expression of *tofI* was lower in the *lon* mutant than in the wild type at the transcription level (Fig 3B). To detect the level of TofI-HA in the wild type and *lon* mutant via western blot using an anti-HA antibody, stationary-phase strains carrying the TofI-HA clone, pTOFI6, were used. The level of TofI-HA in the *lon* mutant was approximately 71.1% of the wild type level (Fig 3C).

## Low levels of oxalate biosynthetic enzymes produced by the *lon* mutant

Because the *lon* mutant produced no detectable QS signal, we measured the capacity for oxalate biosynthesis, which is known to be QS-dependent, in the *lon* mutant. We found that the oxalate level was significantly lower in the *lon* mutant than in the wild type (Fig 4A). Reduced production of oxalate by the *lon* mutant was in accordance with the observed increase in pH of the *lon* mutant (Figs 2C and 4A). To determine whether the cause of the low oxalate level in the *lon* mutant was due to the low expression of *obcA* and *obcB*, which are responsible for oxalate biosynthesis, we measured the expression levels of both genes in the wild type and the *lon* mutant. The expression of the *obcA* and *obcB* genes was significantly lower in the *lon* mutant than in the wild type and the complemented strains (Fig 4B). We also analyzed the levels of ObcA and ObcB by western blotting using ObcA and ObcB antibodies at the time points corresponding to the mid-log, early stationary, and mid-stationary growth phases after subculture (6, 12, and 24 hours, respectively). ObcA and ObcB were not detected in the *lon* mutant at 6 hours but were detected at 12 hours after subculture (Fig 4C). Oxalate production was significantly lower in the *lon* mutant than in the wild type, even though the ObcA and ObcB were accumulated in the *lon* mutant during growth (Fig 4A and 4C). To investigate whether Lon affects the activities of ObcA and ObcB, we measured specific oxalate biosynthetic activity in the wild type and *lon* mutant. Specific oxalate biosynthetic activity was reduced by 68.86% in

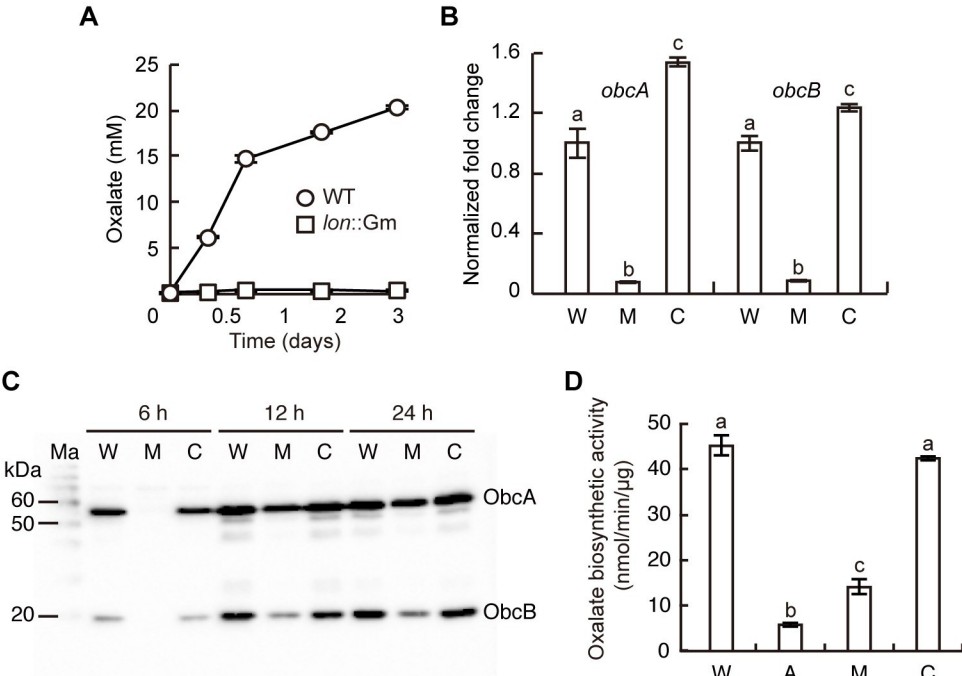

**Fig 4. Reduced production of oxalate by the *lon* mutant of *B. glumae*.** (A) Oxalate in the culture supernatant of each strain was measured at a designated time point. The wild type biosynthesized approximately 20 mM oxalate, whereas the *lon* mutant produced very little oxalate throughout its growth. (B) Expression of *obcA* and *obcB* in each strain during mid-log phase was determined by qRT-PCR. W, M, and C denote the wild type, *lon* mutant, and complemented strains, respectively. The letters (a, b, and c) above each mean indicate significant differences based on a one-way ANOVA, followed by Tukey's post-hoc analysis. A value of $p < 0.05$ indicates significant differences among the strains. (C) Western blot analysis using ObcA and ObcB antibodies at 6, 12, and 24 h after subculture indicated lower levels of ObcA and ObcB in the *lon* mutant than in the wild type. Ma, W, M, and C denote the molecular markers, the wild type, *lon* mutant, and complemented strains, respectively. (D) The specific oxalate biosynthetic activity of each strain during early stationary phase was determined. W, A, M, and C denote the wild type, *obcA* mutant, *lon* mutant, and complemented strains, respectively. The letters (a, b, and c) above each mean represent significant differences based on a one-way ANOVA, followed by Tukey's post-hoc analysis. A value of $p < 0.05$ represented significant differences among strains.

the *lon* mutant compared with the wild type (Fig 4D). When specific oxalate biosynthetic activity was converted into the units per CFU, the oxalate biosynthetic activity was significantly lower in the *lon* mutant than in the wild type (S4 Fig).

## Reduced oxalate biosynthetic enzyme activity in the *lon* mutant was not due to low production of C8-HSL

As *obcA* and *obcB* are expressed in a QS-dependent manner in *B. glumae*, and the *lon* mutant produced no detectable amounts of QS signal, we evaluated whether oxalate biosynthetic capacity can be recovered after exogenous addition of C8-HSL to the *lon* mutant culture medium. When levels of ObcA and ObcB and oxalate biosynthetic activity were measured in the *lon* mutant cells grown in LB with the addition of 10 µM C8-HSL, levels of ObcA and ObcB and oxalate biosynthetic activity in the *lon* mutant did not recover to wild-type levels (Fig 5A and 5B). Oxalate production was monitored during growth in LB buffered with 100 mM HEPES (pH 7.0) and supplemented with up to 10 µM C8-HSL for 2 days after subculture, but the *lon* mutant did not produce oxalate as much as the wild type did (Fig 5C).

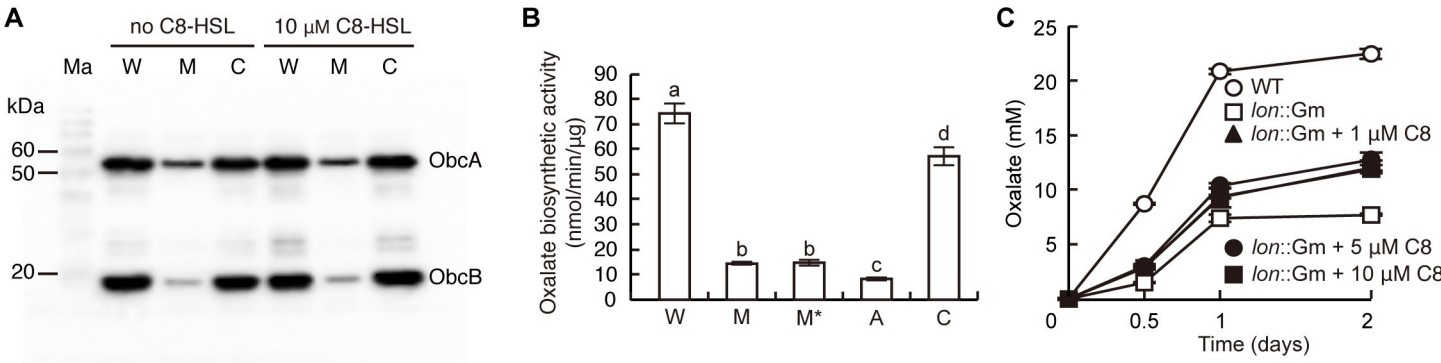

**Fig 5. Supplementation of 10 μM C8-HSL in *lon* mutant cultures did not increase the level of oxalate biosynthetic protein to the wild-type level.** (A) The amount of ObcAB protein in the wild type, *lon* mutant, and complemented strains during early stationary phase was determined via western blot using anti-ObcA and anti-ObcB antibodies. Ma, W, M, and C denote molecular markers, the wild type, *lon* mutant, and complemented strain, respectively. (B) The specific oxalate biosynthetic activity of each strain during early stationary phase was determined. W, M, M*, A, and C denote the wild type, *lon* mutant, *lon* mutant supplemented with 10 μM C8-HSL, *obcA* mutant, and complemented strain, respectively. The letters (a, b, c, and d) above each mean represent significant differences based on a one-way ANOVA followed by Tukey's post-hoc analysis. A value of $p < 0.05$ indicates significant differences among strains. (C) Growth for 2 days after subculture in the LB supplemented with 100 mM HEPES (pH 7.0) and 1 ~ 10 μM C8-HSL did not recover oxalate biosynthesis in the *lon* mutant to the wild type level.

## Influence of Lon protease on the TofR level

As addition of excessive QS signal to the *lon* mutant did not recover the QS-dependent oxalate production to wild-type levels, we hypothesized that Lon protease might influence the activity of the QS signal receptor, TofR. To determine whether Lon protease affects TofR levels, we measured total TofR levels in the *lon* mutant. TofR levels were 1.81- and 1.6-fold higher in the *lon* mutant and the *tofI/lon* double mutant, respectively, than in the wild type (Fig 6A). Although no QS signals were detected in the *tofI* mutant or *lon* mutant, TofR levels in the *tofI* mutant were 12.21% of its wild type and 6.72% of its *lon* mutant (Fig 6A). When *lon* and *tofR* were individually expressed with or without C8-HSL in *E. coli*, C8-HSL did not influence total levels of TofR (Fig 6B). However, TofR levels were reduced by 4.9-fold when the two genes were co-expressed without C8-HSL compared to the individual expression of the *tofR* gene (Fig 6B). Addition of C8-HSL to *E. coli* cultures co-expressing the two genes resulted in a 2.2-fold increase in TofR compared with the same cultures grown without C8-HSL (Fig 6B).

## Interaction of Lon protease with ObcA and ObcB but not with TofI

As the *lon* mutant exhibited low oxalate biosynthetic activity and produced significantly less QS signals compared with the wild type, we hypothesized that Lon protease interacts with ObcA, ObcB, and TofI, affecting their activities. Immunoprecipitation was performed using anti-ObcB and anti-Lon antibodies, ObcA protein was detected, which indicated that ObcA interacts with ObcB and Lon in the wild type strain (Fig 7A). In addition to the 59.11 kDa band of ObcA, another positive band was detected with a size of approximately 52 kDa (Fig 7A). ObcB protein was pulled down using an anti-Lon antibody in an affinity pulldown experiment, which showed that ObcB and Lon interacted in the wild type strain (Fig 7B). To determine whether Lon interacts with TofI, we performed immunoprecipitation using an anti-Lon antibody and detected TofI-HA using an anti-HA antibody by western blot analysis (S5 Fig). TofI-HA was not detected in the immunoprecipitated sample using the anti-Lon antibody (S5 Fig), indicating that Lon and TofI did not interact.

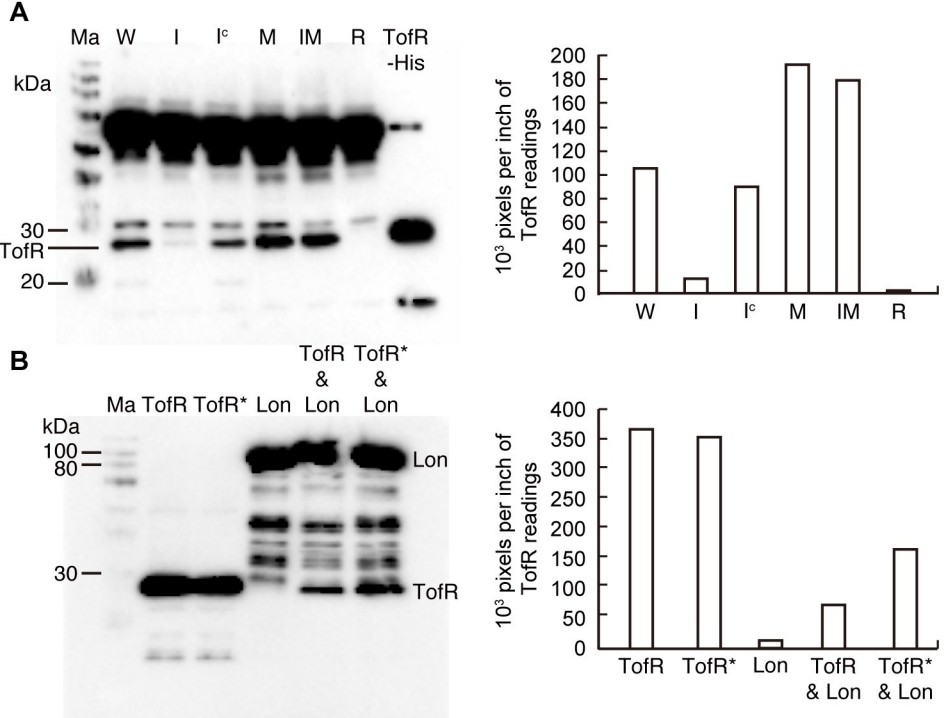

**Fig 6. Lon protease affects the level of TofR.** (A) TofR protein was detected in the *B. glumae* strains grown in the early stationary phase via western blot analysis using an anti-TofR antibody. Ma, W, I, I$^c$, M, IM, R, and TofR-His denote the molecular marker, the wild type, *tofI* mutant, chemically complemented strain of *tofI* mutant with addition of 1 μM C8-HSL, *lon* mutant, *tofI/lon* double mutant, *tofR* mutant, and TofR-His overexpressed in BL21(DE3), respectively. The densities (pixels/inch) of the TofR bands are presented alongside the blot. (B) Western blot analysis using an anti-His antibody showed that Lon influences the levels of TofR, and this influence depends on the presence of C8-HSL. The TofR-His, His-Lon-His, and TofR-His plus His-Lon-His were overexpressed in the BL21(DE3) via addition of 1 mM IPTG with or without 4 μM C8-HSL. The symbols * and & denote addition of 4 μM C8-HSL and co-expression of two proteins, respectively. The densities (pixels/ inch) of TofR bands are shown alongside the blot.

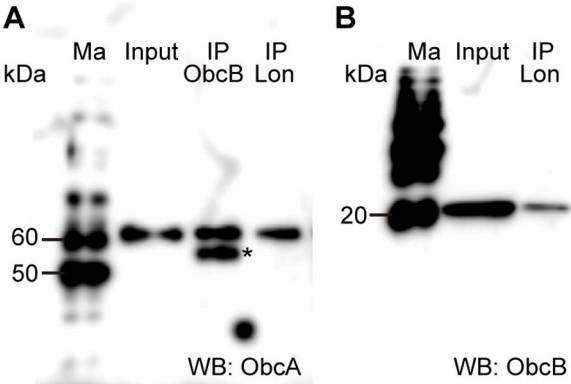

**Fig 7. Lon protease interacted with ObcA and ObcB.** (A) The ObcA protein was pulled down from the total cell lysate of the wild type strain grown in the early stationary phase using anti-ObcB and anti-Lon antibodies in affinity pulldown experiments. Western blotting was performed using an anti-ObcA antibody to detect ObcA in the immunoprecipitated samples. * denote additional unidentified positive signals. (B) The ObcB protein was detected from the total cell lysate of the wild -ype strain grown in the early stationary phase using an anti-Lon antibody. Ma, Input, IP, and WB denote molecular markers, total lysate of the wild type, immunoprecipitation, and western blotting, respectively.

## Discussion

Continuous colony differentiation and abnormal growth due to the loss of Lon protease allowed us to determine that Lon protease is vital to the integrity of cellular functions. It has been reported that the null mutant of *lon* formed large and small colonies in *A. tumefaciens*, but its large colonies were homogeneous after subculture [14]. Although the *lon* mutant has a longer doubling time than that of the wild type *A. tumefaciens*, they reached similar maximum population densities [14]. This study is the first to report that the *lon* mutant colonies are continuously segregated morphologically, and that this mutant cannot grow to its maximum capacity. We explored whether several physiological issues caused by a deficiency of Lon protease threaten bacterial survival *in vitro*.

In general, heterogeneity of the bacterial population is induced by nutrient exhaustion or stress, and it occurs to allow adaptation of the bacteria to environmental changes [23]. Heterogeneous populations may exhibit phenotypic differences within a genetically identical population, or genetic heterogeneity may be present [23]. Persistent colony differentiation was observed in the *lon* mutant of *B. glumae* under all tested culture conditions; therefore, heterogeneity of the *lon* mutant did not appear to be caused by the external environments. The BLONB, generated from BLONN, did not return to the BLONN and was not complemented with pLa4 containing the *lon* gene; other genetic changes appear to have occurred in the BLONB, although we did not identify any mutation in the BLONB genome. The BLONN maintained a homogeneous population only via genetic complementation, indicating that Lon protease plays an essential role in normal growth and cellular integrity.

The primary biological function of Lon protease is control of protein quality by aiding degradation of unfolded proteins and unstable proteins such as global regulators, thereby preventing the accumulation of abnormal proteins [1]. Protein quality control is also important for stress responses; in particular, Lon plays a role in environmental stress responses in *E. coli* [1]. Expression of the *lon* gene in *E. coli* is dependent on the $\sigma^{32}$ transcription factor (RpoH) and is activated by heat shock [24]. These stress-induced proteins are essential for removing damaged polypeptides from stressed cells. High demand for Lon, which suggests that the *lon* gene is upregulated under certain stress conditions, explains the reduced growth of the *lon* mutant under thermal stress [24]. As the promoter region of the *lon* gene in *B. glumae* had no recognition sequence for sigma factors, Lon in *B. glumae* is unlikely to belong to the heat regulon, as observed in *E. coli*.

In addition to proteolytic activity, chaperone-like functions of Lon have also been reported in eukaryotic cells. The human Lon protease LONP1 is a mitochondrial matrix protein that, in conjunction with mitochondrial HSP70, promotes protein folding [25]. Lon plays a vital role in enhancing cell survival by maintaining the stability of the Hsp60–mtHsp70 complex [26]. The interactions of Lon with ObcA and ObcB in *B. glumae* might be due to Lon acting as a chaperone for ObcA and ObcB. The ObcA and ObcB proteins were detected in samples obtained by affinity pulldown using an anti-Lon antibody. Lon clearly interacts with ObcA and ObcB; however, the biochemical mechanism by which Lon functions as a chaperone is not understood and should be addressed in future works. The appearance of additional unidentified positive signals of approximately 52 kDa, as shown in Fig 7A, suggests that ObcA might undergo posttranslational processing. However, the purpose of posttranslational processing of ObcA remains unknown.

The *B. glumae* Lon protease has significant homology with the Lon proteases of other bacteria, and the organization of the *lon* locus is conserved. The locus encoded by *clpX* and *hupB* in the flanking region of *lon* is consistent with those of *P. aeruginosa*, *P. putida*, *E. coli*, and *A. tumefaciens*. The Lon protease of these four different bacteria had 81% (accession

number_AAG05192.1), 67% (AAN67915.1), 70% (AAC73542.1), and 65% (WP_080866091.1) identity with *B. glumae* Lon (bglu_1g13520), respectively (S6 Fig). However, despite these similarities, the function of *B. glumae* Lon is quite different from the functions of Lon in these bacteria. The *P. aeruginosa* Lon negatively controls the LasR/LasI system via degradation of LasI, which directs the synthesis of 3-oxo-dodecanoyl-HSL [15]. Lon negatively regulates the expression of *rhlR/rhlI* via the degradation of LasI [15]. The *lon* mutant of *P. putida* produces significantly higher QS signals than those produced by the wild type [16]. However, in *B. glumae*, no QS signals were detected from the *lon* mutant, even though the TofI level in the *lon* mutant was approximately 71.1% of the wild type level. Lon is unlikely to function as a chaperon for TofI activity, as TofI does not interact with Lon in *B. glumae*. However, Lon might affect other proteins that are involved in AI biosynthesis, such as acyl carrier protein (ACP) or enzymes associated with *S*-adenosylmethionine (SAM). Whether Lon affects the biosynthesis of two substrates of TofI at any level is an interesting topic for future research.

No recovery of the QS-dependent phenotype in the *lon* mutant with excessive addition of C8-HSL QS signal suggests that TofR might not function properly as a receptor protein of C8-HSL in the *lon* mutant. It has been reported that ClpB and Lon protease of *E. coli* are responsible for degradation of TraR, a receptor protein for the QS signal, in *A. tumefaciens* [27]. However, there is no evidence that ClpB or Lon protease of *A. tumefaciens* directly affects the TraR level dependent on the presence of the QS signal. In contrast to TraR of *A. tumefaciens*, we showed that TofR appears to be degraded by the Lon protease of *B. glumae* in *E. coli* when C8-HSL is unavailable. As overexpressed TofR in the absence of C8-HSL was completely insoluble, degradation of TofR by Lon protease *in vitro* was not demonstrable.

Our results revealed previously unidentified roles of Lon protease in bacteria. Continuous segregation and growth defects of the *lon* mutant of *B. glumae* are unique among the phenotypes reported in bacteria. The negative role of Lon protease in bacterial QS is contradictory to the previously known activities of this enzyme in other bacteria. Our findings raise fundamental questions about the functions of bacterial Lon protease, including the mechanism underlying continuous segregation of colony morphology. Our findings suggest that Lon protease plays essential roles in QS and morpho-physiological traits, and that it is a potential target for the development of control agents for the rice pathogen *B. glumae*.

## Supporting information

**S1 Fig. Structural alignment of three homologs of ATP-dependent Lon protease in *B. glumae*.** Domain names were marked in the black text boxes. The e-values and accession numbers below the representing domains were obtained from NCBI Conserved Domains. (PDF)

**S2 Fig. Population density of BGR1, *lon*::Gm, and *lon*::Gm/*lon* grown in LB supplemented with 100 mM HEPES (pH 7.0).** (A) The *lon* mutant had a significantly lower initial growth rate than the wild type, even in buffered LB medium. (B) The population crash of the *lon* mutant was rescued after growing in LB supplemented with 100 mM HEPES (pH 7.0). (PDF)

**S3 Fig. Acyl-homoserine lactone (acyl-HSL) levels were lower in the *lon* mutant than the wild type, even when cells were pooled at $10^9$ CFU/mL and cultured for an additional 24 h in LB supplemented with 100 mM HEPES (pH 7.0).** Thin-layer chromatography (TLC) analysis was performed for acyl-HSLs extracted from the *lon* mutant ($3.57 \times 10^8$ CFU/mL) and the pooled *lon* mutant ($3.57 \times 10^9$ CFU/mL) grown for 24 h in LB supplemented with 100 mM HEPES (pH 7.0). The acyl-HSL sensor strain CV026 was used to visualize C6-HSL and

C8-HSL. W, M, and M2 denote the wild type, *lon* mutant, and pooled *lon* mutant, respectively.
(PDF)

**S4 Fig. Oxalate biosynthetic component activity (units per CFU).** The letters (a, b, and c) above each mean represent significant differences based on a one-way analysis of variance (ANOVA), followed by Tukey's post-hoc analysis. A value of $p < 0.05$ indicates significant differences among strains.
(PDF)

**S5 Fig. No detection of TofI-HA in the immunoprecipitated sample using an anti-Lon antibody.** Strain S2HA carrying the TofI-HA clone, pTOFI6, was used to determine interaction between Lon and TofI.
(PDF)

**S6 Fig. Similarities among ATP-dependent Lon proteases of *B. glumae*, *Pseudomonas aeruginosa*, *Pseudomonas putida*, *Escherichia coli*, and *Agrobacterium tumefaciens*.** Comparison between ATP-dependent Lon proteases of *B. glumae* BGR1 (bglu_1g13520), *P. aeruginosa* PAO1 (AAG05192.1), *P. putida* KT2440 (AAN67065.1), *E. coli* K-12 substr. MG1655 (AAC73542.1), and *A. tumefaciens* C58 (WP_080866091.1).
(PDF)

**S1 Table. The strains and plasmids used in this study.**
(DOCX)

**S2 Table. The primers used in this study.**
(DOCX)

## Author Contributions

**Conceptualization:** Eunhye Goo, Ingyu Hwang.

**Data curation:** Eunhye Goo.

**Funding acquisition:** Eunhye Goo.

**Investigation:** Eunhye Goo, Ingyu Hwang.

**Writing – original draft:** Eunhye Goo, Ingyu Hwang.

**Writing – review & editing:** Eunhye Goo, Ingyu Hwang.

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
