## [Decision Letter · Decision Letter 0]

9 Aug 2021

PONE-D-21-24187

Essential roles of Lon protease in the morpho-physiological traits of the rice pathogen *Burkholderia glumae*

PLOS ONE

Dear Dr. Eunhye Goo,

Thank you for submitting your manuscript to PLOS ONE. After careful consideration, we feel that it has merit but does not fully meet PLOS ONE’s publication criteria as it currently stands. Therefore, we invite you to submit a revised version of the manuscript that addresses the points raised during the review process.

We look forward to receiving your revised manuscript.

Kind regards,

Ya-Wen He, Ph.D

Academic Editor

PLOS ONE

Journal Requirements:

Reviewers' comments:

Reviewer's Responses to Questions

**Comments to the Author**

1. Is the manuscript technically sound, and do the data support the conclusions?

Reviewer #1: Yes

Reviewer #2: Yes

2. Has the statistical analysis been performed appropriately and rigorously? 

Reviewer #1: Yes

Reviewer #2: Yes

3. Have the authors made all data underlying the findings in their manuscript fully available?

Reviewer #1: Yes

Reviewer #2: Yes

4. Is the manuscript presented in an intelligible fashion and written in standard English?

Reviewer #1: Yes

Reviewer #2: Yes

5. Review Comments to the Author

Reviewer #1: Comments to the Author

Summary:

This paper by Eunhye Goo* & Ingyu H describes experiments to decipher the roles of Lon protease for morpho-physiological traits, including colony morphology, growth, QS system, and oxalate biosynthesis in the rice pathogen Burkholderia glumae.

Overall I think that the study provides some interesting insights, but there are some areas in the manuscript where the authors should clarify to help the reader understand. Further follow up and some clarifications would strengthen the paper.

Major comments are detailed below.

1. In the methods section, “Measurements of oxalate levels and oxalate biosynthetic activity assay” and “Autoinducer assay”, the authors should briefly describe how you performed these experiments in this study.

2. In the results section, the authors should indicate how you came to this conclusion shown in Line 174-180. In addition, please consider if you need a figure to show the structural alignment of these homologous proteins.

3. Line 189-191, The authors state: “Such segregation of the BLONB and BLONN types did not change in M9 minimal medium supplemented with 0.2% glucose (Fig. 1E), nutrient broth medium (Fig. 1F), King’s B medium (Fig. 1G), and LB supplemented with 100mM HEPES (pH 7.0) (Fig. 1H).” Please make sure whether these figures are complete. And looking at these figures, the results do not appear to be consistent. How do the authors explain the obvious difference between Fig. 1A and 1E-1F, or between Fig. 1B and 1E-1F? Please indicate how you distinguished between BLONN and BLONB in Fig. 1A, 1B, and 1E-1H, since it’s a a little hard for readers to judge.

4. After diving thorough the Methods section, I did not find out what are the relevant approaches used in the measurement of cell viability and extracellular pH assay. Please help the reader to understand the assay. There is an obvious difference of the initial bacteria density shown in Fig. 2A and Fig. 2B. So, when exactly were the samples taken for analysis? In addition, the font and axis value should be consistent.

5. Line 204-205, The authors state: “Environmental pH increased during growth of the lon mutant, which resulted in a population crash (Fig. 2).” Please clarify this statement, since I don’t see the results of “a population crash” from Fig. 2.

6. Line 251-257, the description of the results is too general to highlight the work shown in Fig. 5A-C. Please describe the results objectively to help the reader help to understand how you get the conclusion.

7. In the manuscript, hypotheses are proposed in Line 222 and 260 lacking a proper information to understand the point of these experiment. It would be helpful for readers to indicate the function of tofI and TofR and the correlation between tofI or TofR and Lon protease in the introduction or the corresponding result section.

8. Line 314-316, the authors state: “As the promoter region of the lon gene in B. glumae had no recognition sequence for sigma factors, Lon in B. glumae is unlikely to belong to the heat regulon, as observed in E. coli.” Is there any evidence of this? As pointed out in Line 312-313, “the lon gene is upregulated under certain stress conditions”. So did the authors test the expression of lon gene under heat shock?

9. Line 314-316, this statement of the similarities of the Lon protease among four different bacteria lacks proper presentation of alignment results.

Mirror comments:

1. Figure legends should briefly describe the technique and/or experimental design used. The result section can be unnecessary.

2. There are few recent studies in the references. Please consider update references of discuss section.

3. Please unify the primers’ name in Line 105 ( “Tp or TP”).

4. Line 191, “100mM” should be “100 mM”.

Reviewer #2: Reviewing comments for PONE-D-21-24187

Lon protease plays a negative regulatory role in the quorum sensing (QS) system of P. aeruginosa and P. putida but the mechanisms remain unknown. This study wants to investigate the role of Lon protease in growth capacity, failed QS signal biosynthesis and colony size variation etc.

I have the following comments.

• Major concerns

1) In Results part (row 172-180), can you use a supplementary figure to explain the difference and show this diagram among 13520, 31260 and 33380?

2) For Figure 1A, what medium did you use? From Figure 1B, why there is no BLONN but only BLONB and BLONS were found from BLONB? Have you tried to grow BLONB in different medium and same results were obtained?

3) For Figure 2B and 2C, how about the cell number and pH in complemented strain?

4) For Figure 3A, strong signal can be detected in wild type when the number can be more than 109 (left). However, the signal is quite weak for the wild type on the right part when the strain number is 3.48 x 109. Why did they show such significant difference? Similarly, the signal is also very weak for complemented strain when the number is more than 3.03 x 109 (right side on Figure 3A). Is this caused by different exposing time or others?

5) For the ‘failure of QS signal biosynthesis in the lon mutant (row 215-227), can the author make a conclusion for the analysis?

6) In Figure 4D, the author has checked the oxalate biosynthetic activity in obcA mutant. How about the activity in obcB mutant? Similar as obcA?

7) In figure 5, we can conclude that adding with C8-HSL can not rescue the reduced oxalate biosynthetic enzyme activity. Have you ever tried C6-HSL?

8) For Figure 7A, is the additional signal band produced by phosphorylation?

9) In Figure 2S, how about the HSL level if we extend the time to 48h or even 72 hrs?

10) What’s the relationship between tofI and tofR? Can the author add some literatures to explain in the main text to benefit the understanding? What’s the relationship between tofI and QS signal? Please also cite a paper to briefly explain it in the result part (row 221).

• Minor concerbs

Row mistakes or errors suggestions

86 Gm in the main text and figures, ‘gm’ is used. Please use the same word in the manuscript.

222 tofI and tofR What are their full names?

279-282 Revise the sentence

388, 396 etc ‘available from….’ Pls use the same format for all cited papers.

465-475 BLONB etc denote the B-Big, N-normal and S-small size in the figure legend

486 produced than consider to revise the sentence here

535 the * and & both of them denote the addition of 4uM C8-HSL?

38 AHL if possible, pls use HSL in the main text

6. PLOS authors have the option to publish the peer review history of their article (what does this mean?). If published, this will include your full peer review and any attached files.

Reviewer #1: No

Reviewer #2: **Yes: **Gu Keyu

---

## [Author Response · Author response to Decision Letter 0]

22 Aug 2021

Reviewer #1: Comments to the Author

Summary:

This paper by Eunhye Goo* & Ingyu Hwang describes experiments to decipher the roles of Lon protease for morpho-physiological traits, including colony morphology, growth, QS system, and oxalate biosynthesis in the rice pathogen Burkholderia glumae.

Overall I think that the study provides some interesting insights, but there are some areas in the manuscript where the authors should clarify to help the reader understand. Further follow up and some clarifications would strengthen the paper.

Major comments are detailed below.

1. In the methods section, “Measurements of oxalate levels and oxalate biosynthetic activity assay” and “Autoinducer assay”, the authors should briefly describe how you performed these experiments in this study.

- Oxalate was measured using an oxalate assay kit (Trinity Biotech) according to the manufacturer’s instructions. Briefly, oxalate was converted into carbon dioxide and hydrogen peroxide using oxalate oxidase, and the production of hydrogen peroxide was measured via reaction with 3-(dimethylamino) benzoic acid, during which it formed a blue compound catalyzed by peroxidase. Absorbance at 590 nm was measured using a microplate reader (PerkinElmer); 0.5 mM oxalate was used as a standard. The absorbance of the sample was divided by the absorbance of standard and multiplied by the appropriate dilution factor. The reaction buffer used in the assay measuring oxalate biosynthetic activity consisted of 200 mM Tris-Cl (pH 8.0), 10 mM EDTA, 20 mM CoCl2, 2 mM acetyl-CoA, and 200 mM oxaloacetate. The total cell lysate was added to the reaction mixture, followed by incubation at 37°C for 10 min. The level of biosynthesized oxalate from the reaction was measured using an oxalate assay kit. Units were calculated as the concentration of biosynthesized oxalate divided by the amount of total protein and reaction time. Autoinducer was extracted from the 500 µL culture-free supernatant using the same amount of ethyl acetate and dried. The dried pellet was dissolved in 10 µL dimethyl sulfoxide. Autoinducer was developed via the thin layer chromatography (TLC), using 70 % methanol in the TLC tank. LB agar with biosensor strain Chromobacterium violaceum CV026 overlaid was used to develop the TLC plate to visualize C8-HSL and C6-HSL. We have added a description to the Methods section.

2. In the results section, the authors should indicate how you came to this conclusion shown in Line 205-211. In addition, please consider if you need a figure to show the structural alignment of these homologous proteins.

- We have added a Supplementary Figure (S1 Fig) to show the structural alignment of three homologs of ATP-dependent Lon protease in B. glumae.

3. Line 222-224, The authors state: “Such segregation of the BLONB and BLONN types did not change in M9 minimal medium supplemented with 0.2% glucose (Fig. 1E), nutrient broth medium (Fig. 1F), King’s B medium (Fig. 1G), and LB supplemented with 100mM HEPES (pH 7.0) (Fig. 1H).” Please make sure whether these figures are complete. And looking at these figures, the results do not appear to be consistent. How do the authors explain the obvious difference between Fig. 1A and 1E-1F, or between Fig. 1B and 1E-1F? Please indicate how you distinguished between BLONN and BLONB in Fig. 1A, 1B, and 1E-1H, since it’s a little hard for readers to judge.

- Colony differentiation was consistently showed in M9 minimal medium supplemented with 0.2 % glucose (Fig. 1E), nutrient broth medium (Fig. 1F), King’s B medium (Fig. 1G), and LB supplemented with 100 mM HEPES (pH 7.0) (Fig. 1H). BLONB was bigger than BLONN in all tested media, and was darker, than BLONN under a dissecting microscope. Fig. 1A and 1E–1H were obtained from BLONN subcultured in different media. Fig. 1B shows that the BLONB segregated into BLONB and BLONS only, not BLONN, which did not change in all tested media (data now shown). We have added an explanations on lines 216–218. 

4. After diving thorough the Methods section, I did not find out what are the relevant approaches used in the measurement of cell viability and extracellular pH assay. Please help the reader to understand the assay. There is an obvious difference of the initial bacteria density shown in Fig. 2A and Fig. 2B. So, when exactly were the samples taken for analysis? In addition, the font and axis value should be consistent.

- Cells were inoculated in 2 mL LB broth with appropriate antibiotics and grown at 37°C at 250 rpm for 18 h. Overnight cultures were washed twice with fresh LB broth, and turbidity was adjusted to an optical density (OD) of 0.05 at 600 nm using a BioSpectrometer (Eppendorf) followed by 2 mL subculture in each of glass test tubes (PYREX) for all assays. To compare initial growth rate of wild type and mutant, an OD600 of 0.05 was diluted 20 times followed by subculture (Figs. 2A and S1A Fig). Aliquots of 100 µL from each sample were serially diluted and 10 µL each of three repeats was spotted on LB agar medium to monitor colony-forming units (CFUs) at the designated time point. LB agar plates were incubated at 37°C for 24 h to allow colonies to grow. CFUs were counted under a dissecting microscope and multiplied by the appropriated dilution factor to calculate CFU/mL. In an extracellular pH assay, the culture supernatant was sampled from each vial at the designated time point and the pH was measured using a pH meter (Lab 860, SCOTT Instruments). We have added a description of the measurement of cell viability and the extracellular pH assay in the Methods section. Although we adjusted the optical densities of overnight cultures to approximately similar cell densities for growth studies, there remains some differences in the initial cell numbers. However, we believe that these discrepancies do not affect our growth studies because all cells were at late stationary phase when they were diluted and subcultured. To clearly visualize the biological significance, we used different axis values in Fig. 2A and Fig. 2B, but the same font size is consistently used.

5. Line 237-238, The authors state: “Environmental pH increased during growth of the lon mutant, which resulted in a population crash (Fig. 2).” Please clarify this statement, since I don’t see the results of “a population crash” from Fig. 2.

- The population of the lon mutant gradually decreased after 12 hours of subculture, and viable cells were not detected after 3 days of subculture. As per your advice, we have added the results of the population crash at 4 days after subculture in Fig. 2B.

6. Line 285-290, the description of the results is too general to highlight the work shown in Fig. 5A-C. Please describe the results objectively to help the reader help to understand how you get the conclusion.

There were no differences in the levels of ObcA and ObcB proteins in the lon mutant between no C8-HSL and supplementation with 10 µM C8-HSL, although the gene expression of obcA and obcB is dependent on the C8-HSL-mediated QS system in B. glumae. The oxalate biosynthetic activity of the lon mutant was not increased with the addition of 10 µM C8-HSL. Even with supplementation of C8-HSL up to 10 µM, oxalate production was not recovered to the wild-type levels in lon mutant cells grown in LB buffered with 100 mM HEPES (pH 7.0). We have expanded our description of these results in the revised manuscript. 

7. In the manuscript, hypotheses are proposed in Line 254-255 and 296-297 lacking a proper information to understand the point of these experiment. It would be helpful for readers to indicate the function of tofI and TofR and the correlation between tofI or TofR and Lon protease in the introduction or the corresponding result section.

- B. glumae possesses a LuxI/R type QS system, which uses C8-HSL synthesized by TofI, QS signal synthase, and its cognate receptor TofR to regulate the expression of QS-dependent genes. We determined whether the lack of QS signals in the lon mutant was due to the low expression of the tofI gene (line 254-255). We have added additional information in the Results section to help readers understand the rationale of these experiments.

8. Line 350-352, the authors state: “As the promoter region of the lon gene in B. glumae had no recognition sequence for sigma factors, Lon in B. glumae is unlikely to belong to the heat regulon, as observed in E. coli.” Is there any evidence of this? As pointed out in Line 348-349, “the lon gene is upregulated under certain stress conditions”. So did the authors test the expression of lon gene under heat shock?

- The recognition sequence of sigma-70, sigma-32, and sigma-54 in E. coli has been previously reported (reference 24 in manuscript). There was no recognition sequence of sigma-70, sigma-32, and sigma-54 in the promoter region of the lon gene in B. glumae. Expression of the lon gene in E. coli is activated by heat shock (reference 24 in manuscript). We tested the expression of Lon in the wild-type of B. glumae under heat stress conditions via Western blotting using an anti-Lon antibody, but it was not upregulated by heat shock (data now shown).

9. Line 368-370, this statement of the similarities of the Lon protease among four different bacteria lacks proper presentation of alignment results.

- We have added a Supplementary Figure (S6 Fig) to present the structural alignment of Lon protease homologs of four different bacteria.

Mirror comments:

1. Figure legends should briefly describe the technique and/or experimental design used. The result section can be unnecessary.

- Amended

2. There are few recent studies in the references. Please consider update references of discuss section.

- Amended

3. Please unify the primers’ name in Line 118 ( “Tp or TP”).

- We have changed TP to Tp.

4. Line 224, “100mM” should be “100 mM”.

- Amended

Reviewer #2: Reviewing comments for PONE-D-21-24187

Lon protease plays a negative regulatory role in the quorum sensing (QS) system of P. aeruginosa and P. putida but the mechanisms remain unknown. This study wants to investigate the role of Lon protease in growth capacity, failed QS signal biosynthesis and colony size variation etc.

I have the following comments.

• Major concerns

1) In Results part (row 203-211), can you use a supplementary figure to explain the difference and show this diagram among 13520, 31260 and 33380?

- We have added a Supplementary Figure (S1 Fig) to show the structural alignment of three homologs of ATP-dependent Lon protease in B. glumae.

2) For Figure 1A, what medium did you use? From Figure 1B, why there is no BLONN but only BLONB and BLONS were found from BLONB? Have you tried to grow BLONB in different medium and same results were obtained?

- LB medium was used for Figure 1A. This information has been added to the figure legend. The BLONB was segregated into BLONB and BLONS only, not BLONN, which did not change in all tested media (data now shown).

3) For Figure 2B and 2C, how about the cell number and pH in complemented strain?

- The complemented strain showed the same patterns of cell population and extracellular pH as the wild type.

4) For Figure 3A, strong signal can be detected in wild type when the number can be more than 109 (left). However, the signal is quite weak for the wild type on the right part when the strain number is 3.48 x 109. Why did they show such significant difference? Similarly, the signal is also very weak for complemented strain when the number is more than 3.03 x 109 (right side on Figure 3A). Is this caused by different exposing time or others?

- The visualizing signals in an autoinducer assay can depend on the loading amount of extracted samples and cell number of biosensor Chromobacterium violaceum CV026. Because the left and right sides of Fig. 3A were determined with different TLC plates, it is difficult to compare them directly.

5) For the ‘failure of QS signal biosynthesis in the lon mutant (row 247-260), can the author make a conclusion for the analysis?

- QS signal biosynthesis is closely related to cell density. Because the cell density of the lon mutant does not rise to the maximum population, pooled samples were then further cultured to see if the QS signal was biosynthesized, but the QS signal was not detected. Lon protease can affect other proteins involved in QS signal biosynthesis because QS signal synthase TofI was detected in the lon mutant. These comments have been included in the Discussion.

6) In Figure 4D, the author has checked the oxalate biosynthetic activity in obcA mutant. How about the activity in obcB mutant? Similar as obcA?

- The obcB mutant did not produce oxalate (reference 20 in manuscript). Oxalate biosynthetic activity in the obcB mutant might be similar to that of the obcA mutant.

7) In figure 5, we can conclude that adding with C8-HSL can not rescue the reduced oxalate biosynthetic enzyme activity. Have you ever tried C6-HSL?

- The obcA and obcB gene expression is dependent on the C8-HSL-mediated QS system in B. glumae. The addition of C6-HSL to the lon mutant culture did not change any phenotypes, including the oxalate biosynthetic enzyme activity and growth.

8) For Figure 7A, is the additional signal band produced by phosphorylation?

- We did not obtain direct evidence of an additional positive signal produced by phosphorylation, but ObcA may be phosphorylated based on our phosphorylation prediction results.

9) In Figure 2S, how about the HSL level if we extend the time to 48h or even 72 hrs?

- Autoinducer was not detected in the lon mutant grown in LB supplemented with HEPES (pH 7.0) for 72 hours. We expected that the same results might be reproduced from pooled cells of the lon mutant grown for 72 hours.

10) What’s the relationship between tofI and tofR? Can the author add some literatures to explain in the main text to benefit the understanding? What’s the relationship between tofI and QS signal? Please also cite a paper to briefly explain it in the result part (row 254-255).

- B. glumae uses C8-HSL synthesized by TofI, QS signal synthase, and its cognate receptor TofR to regulate the expression of QS-dependent genes. We determined whether the lack of QS signals in the lon mutant was due to the low expression of the tofI gene (line 254-255). We have added additional information in the Results section to help readers understand the rationale of these experiments.

• Minor concerns

Row mistakes or errors suggestions

99 Gm in the main text and figures, ‘gm’ is used. Please use the same word in the manuscript.

- Amended

254-255 tofI and tofR What are their full names?

- The tofI is a QS signal synthase, and tofR is the QS signal receptor in B. glumae. We have added this information to the Results section.

315-317 Revise the sentence

- We have revised this sentence accordingly.

429, 439 etc ‘available from….’ Pls use the same format for all cited papers.

- Amended

524-534 BLONB etc denote the B-Big, N-normal and S-small size in the figure legend

- Amended

545 produced than consider to revise the sentence here

- Amended

597 the * and & both of them denote the addition of 4uM C8-HSL?

- The symbol * denotes the addition of 4 µM C8-HSL, and the symbol & denotes co-expression of Lon and TofR proteins. For example, TofR* & Lon indicates co-expression of TofR and Lon supplemented with 4 µM C8-HSL.

547 AHL if possible, pls use HSL in the main text

- We have changed AHL to acyl-HSL.

---

## [Decision Letter · Decision Letter 1]

27 Aug 2021

Essential roles of Lon protease in the morpho-physiological traits of the rice pathogen *Burkholderia glumae*

PONE-D-21-24187R1

Dear Dr. Goo,

We’re pleased to inform you that your manuscript has been judged scientifically suitable for publication and will be formally accepted for publication once it meets all outstanding technical requirements.

Kind regards,

Ya-Wen He, Ph.D

Academic Editor

PLOS ONE

Additional Editor Comments (optional):

Reviewers' comments:

Reviewer's Responses to Questions

**Comments to the Author**

1. If the authors have adequately addressed your comments raised in a previous round of review and you feel that this manuscript is now acceptable for publication, you may indicate that here to bypass the “Comments to the Author” section, enter your conflict of interest statement in the “Confidential to Editor” section, and submit your "Accept" recommendation.

Reviewer #2: All comments have been addressed

2. Is the manuscript technically sound, and do the data support the conclusions?

Reviewer #2: Yes

3. Has the statistical analysis been performed appropriately and rigorously? 

Reviewer #2: Yes

4. Have the authors made all data underlying the findings in their manuscript fully available?

Reviewer #2: Yes

5. Is the manuscript presented in an intelligible fashion and written in standard English?

Reviewer #2: Yes

6. Review Comments to the Author

Reviewer #2: Thanks the authors to revise it and explain it in details. This version is much readable now. I strongly recommend for publishing in this journal.

7. PLOS authors have the option to publish the peer review history of their article (what does this mean?). If published, this will include your full peer review and any attached files.

Reviewer #2: **Yes: **Gu Keyu

---

## [Editor Report · Acceptance letter]

2 Sep 2021

PONE-D-21-24187R1 

Essential roles of Lon protease in the morpho-physiological traits of the rice pathogen *Burkholderia glumae*

Dear Dr. Goo:

I'm pleased to inform you that your manuscript has been deemed suitable for publication in PLOS ONE. Congratulations! Your manuscript is now with our production department. 

Kind regards, 

on behalf of

Dr Ya-Wen He 

Academic Editor

PLOS ONE